# Privacy-Preserving Data Filtering in Federated Learning Using Influence Approximation

## Abstract

Federated Learning by nature is susceptible to low-quality, corrupted, or even malicious data that can severely degrade the quality of the learned model. Traditional techniques for data valuation cannot be applied as the data is never revealed. We present a novel technique for filtering, and scoring data based on a *practical influence approximation* ('lazy' influence) that can be implemented in a *privacy-preserving* manner. Each agent uses his *own data* to evaluate the influence of another agent's batch, and reports to the center an obfuscated score using differential privacy. Our technique allows for highly effective filtering of corrupted data in a variety of applications. Importantly, the accuracy does not degrade significantly, even under really strong privacy guarantees ($\varepsilon \leq 1$), especially under realistic percentages of mislabeled data.

## 1 Introduction

The success of Machine Learning (ML) depends to a large extent on the availability of high-quality data. This is a particularly important issue in Federated Learning (FL) since the model is trained without access to raw training data. Instead, a single *center* uses data held by a set of independent and sometimes self-interested *data holders* to jointly train a model. Having the ability to *score* and *filter* irrelevant, noisy, or malicious data can (i) significantly improve model accuracy, (ii) speed up training, and even (iii) reduce costs for the center when it pays for data.

> We are the *first* to introduce a *practical* approach for *scoring, and filtering* contributed data in a Federated Learning setting that ensures *strong, worst-case privacy*.

A clean way of quantifying the effect of data point(s) on the accuracy of a model is via the notion of *influence* [20, 4]. Intuitively, influence quantifies the marginal contribution of a data point (or batch of points) on a model's accuracy. One can compute this by comparing the difference in the model's empirical risk when trained with and without the point in question. While the influence metric can be highly informative, it is impractical to compute: re-training a model is time-consuming, costly, and often impossible, as agents do not have access to the entire dataset. We propose a simple and practical approximation of the sign of the exact influence (*'lazy' influence approximation*), which is based on an estimate of the direction of the model after a small number of local training epochs with the new data.

Another challenges is to approximate the influence while preserving the privacy of the data. Many approaches to Federated Learning (e.g., [27, 30]) remedy this by combining FL with Differential Privacy (DP) [8, 9, 10, 11], a data anonymization technique that is viewed by many researchers as the gold standard [29]. We show how the sign of influence can be approximated in an FL setting while maintaining strong differential privacy guarantees.

Submitted to 36th Conference on Neural Information Processing Systems (NeurIPS 2022). Do not distribute.

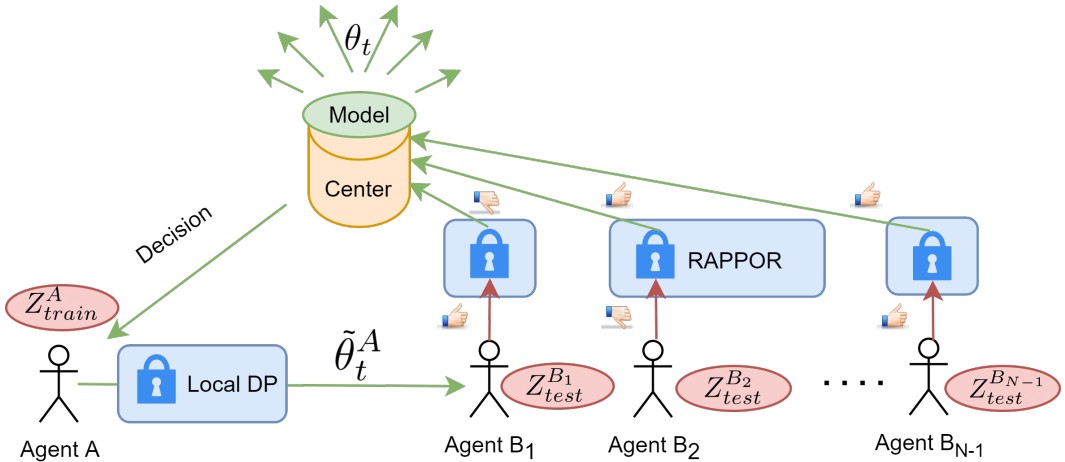

Figure 1: Data filtering procedure. See Section 1.1.

## 1.1 High Level Description of Our Setting

A center $C$ coordinates a set of agents to train a single model (Figure 1). $C$ has a small set of 'warm-up' data which are used to train an initial model $M_0$ that captures the desired input/output relation. We assume that each data holder has a set of training points that will be used to improve the model, and a set of test points that will be used to evaluate the contributions of other agents. To prohibit agents from tailoring their contributions to the test data, it must be kept private. For each federated learning round $t$ (model $M_t$), each data holder agent will assume two roles: the role of the contributor ($A$), and the role of the tester ($B$). As a contributor, an agent performs a small number of local epochs to $M_t$ – enough to get an estimate of the gradient[1] – using a batch of his training data $z_{A,t}$. Subsequently, $A$ sends the updated partial model, with specifically crafted noise, $M_{t,A}$ to every other agent (which assumes the role of a tester). The noise applied protects the update gradient, while still retaining information on the usefulness of data. Each tester $B$ uses its test dataset to approximate the empirical risk of $A$'s training batch (i.e., the approximate influence). This is done by evaluating each test point and comparing the loss. In a FL setting, we can not re-train the model to compute the exact influence; instead, $B$ performs only a small number of training epochs, enough to estimate the direction of the model ('lazy' influence approximation). As such, we opt to look at the sign of the approximate influence (and not the magnitude). Each tester aggregates the signs of the influence for each test point, applies controlled noise, and sends this information to the center. Finally, the center decides to accept $A$'s training batch if the majority of $B$s report positive influence, and reject otherwise.

## 2 Related Work and Discussion

**Federated Learning**    Federated Learning (FL) [25, 19, 32, 22] has emerged as an alternative method to train ML models on data obtained by many different agents. In FL a center coordinates agents who acquire data and provide model updates. FL has been receiving increasing attention in both academia [23, 35, 16, 1] and industry [15, 2], with a plethora of real-world applications (e.g., training models from smartphone data, IoT devices, sensors, etc.).

**Influence functions**    Influence functions are a standard method from robust statistics [4] (see also Section 3), which were recently used as a method of explaining the predictions of black-box models [20]. They have also been used in the context of fast cross-validation in kernel methods and model robustness [24, 3]. While a powerful tool, computing the influence involves too much computation and communication, and it requires access to the train and test data (see [20] and Section 3).

---

[1]The number of local epochs is a hyperparameter. We do not need to fully train the model. See Section 3.2.

**Data Filtering** A common but computationally expensive approach for filtering in ML is to use the Shapley Value of the Influence to evaluate the quality of data [18, 14, 17]. Other work includes for example rule based filtering of least influential points [28], or constructing weighted data subsets (corsets) [5]. While data filtering might not always pose a significant problem in traditional ML, in a FL setting it is more important because even a small percentage of mislabeled data can result in a significant drop in the combined model's accuracy. Moreover, because of the privacy requirements, contributed data is not directly accessible for assessing its quality. [31] propose a decentralized filtering process specific to federated learning, yet they do not provide any formal privacy guarantees. To the best of our knowledge, we are the *first* to provide a *practical* application of influence metrics as a filtering and scoring mechanism for FL that also ensures strong, worst-case Differential Privacy guarantees.

**Differential Privacy** Differential Privacy (DP) [8, 9, 10, 11] has emerged as the de facto standard for protecting the privacy of individuals. Informally, DP captures the increased risk to an individual's privacy incurred by his participation in the learning process. As a simplified intuitive example, consider an agent being surveyed on a sensitive topic. In order to achieve differential privacy, one needs a source of randomness, thus the agent decides to flip a coin. Depending on the result (heads or tails), an agent can reply truthfully, or at random. Now an attacker can not know if the decision was taken based on the agent's actual preference, or due to the coin toss. Of course, to get meaningful results, we need to bias the coin towards the true data. In this simple example, the logarithm of the ratio $Pr[\text{heads}]/Pr[\text{tails}]$ represent the privacy cost (also referred to as the privacy budget), denoted traditionally by $\varepsilon$. For a more comprehensive overview, we refer the reader to [29, 12].

# 3 Methodology

We aim to address two challenges: approximating the influence of a (batch of) datapoint(s) without having to re-train the entire model from scratch, and protecting the privacy of both the train and test dataset of each agent. This is important not only to protect the sensitive information of users, but also to ensure that malicious agents can not tailor their contributions to the test data. We first introduce the notion of *influence* [4], and our approach to approximating this value. Second, we describe a differentially private reporting scheme for crowdsourcing the approximate influence values from the testers.

We consider a classification problem from some input space $\mathcal{X}$ (e.g., features, images, etc.) to an output space $\mathcal{Y}$ (e.g., labels). In a Federated Learning setting, there is a center $C$ that wants to learn a model $M(\theta)$ parameterized by $\theta \in \Theta$, with a non-negative loss function $L(z, \theta)$ on a sample $z = (\bar{x}, y) \in \mathcal{X} \times \mathcal{Y}$. Let $R(Z, \theta) = \frac{1}{n} \sum_{i=1}^{n} L(z_i, \theta)$ denote the empirical risk, given a set of data $Z = \{z_i\}_{i=1}^{n}$. We assume that the empirical risk is differentiable in $\theta$. The training data are supplied by a set of data holders.

## 3.1 Exact Influence

In simple terms, influence measures the marginal contribution of a data point on a model's accuracy. A positive influence value indicates that a data point improves model accuracy, and vice-versa. More specifically, let $Z = \{z_i\}_{i=1}^{n}$, $Z_{+j} = Z \cup z_j$ where $z_j \notin Z$, and let

$$\hat{R} = \min_{\theta} R(Z, \theta) \quad \text{and} \quad \hat{R}_{+j} = \min_{\theta} R(Z_{+j}, \theta)$$

i.e., $\hat{R}$ and $\hat{R}_{+j}$ denote the minimum empirical risk their respective set of data. The *influence* of datapoint $z_j$ on $Z$ is defined as:

$$\mathcal{I}(z_j, Z) \triangleq \hat{R} - \hat{R}_{+j} \tag{1}$$

Despite being highly informative, influence functions have not achieved widespread use in Federated Learning (or Machine Learning in general). This is mainly due to the computational cost. Equation 1 requires a complete retrain of the model, which is time-consuming, and very costly; especially for state-of-the-art, large ML models. Moreover, specifically in our setting, we do not have direct access to the training data. In the following section, we will introduce a practical approximation of the influence, applicable in Federated Learning scenarios.

### 3.2 'Lazy' Influence: A Practical Influence Metric for Filtering Data in FL Applications

The key idea is that *we do not need to approximate the influence value* to filter data; we only need an accurate estimate of its *sign* (in expectation). Recall that a positive influence value indicates that a data point improves model accuracy, and vice-versa, thus we only need to approximate the sign of Equation 1, and use that information to **filter out data with *negative sign***.

Our proposed approach works as follows (recall that each data holder agent assumes two roles: the role of the contributor ($A$), and the role of the tester ($B$)):

  **(i)** For each federated learning round $t$ (model $M_t(\theta_t)$), the contributor agent $A$ performs a small number $k$ of local epochs to $M_t$ using a batch of his training data $Z_{A,t}$, resulting in $\tilde{\theta}_t^A$. $k$ is a hyperparameter. $\tilde{\theta}_t^A$ is the partially trained model of Agent $A$, where most of the layers, except the last one have been frozen. The model should not be fully trained for three key reasons: efficiency, avoiding over-fitting, and preventing the testers ($B$s) from acquiring agent $A$'s model update (e.g., in our simulations we only performed 1 epoch). Furthermore, Agent $A$ adds precise noise to the trained parameters, to ensure strong, worst-case differential privacy. Specifically, Gaussian noise, parametrized by $\sigma$ and a clipping threshold, is added by Agent $A$ to their partial model update, based on [26]. Finally, $A$ sends $\tilde{\theta}_t^A$ to every other agent.

  **(ii)** Each tester $B$ uses his test dataset $Z_{test}^B$ to estimate the sign of the influence using Equation 2. Next, the tester applies noise to $I_{proposed}(Z_{test}^B)$, as will be explained in the next section, to ensure strong, worst-case differential privacy guarantees (i.e., keep his test dataset private).

$$\mathcal{I}_{proposed}(Z_{test}^B) \triangleq \text{sign}\left(\sum_{z_{test}\in Z_{test}^B} L(z_{test}, \theta_t) - L(z_{test}, \theta_t^A)\right) \tag{2}$$

  **(iii)** Finally, the center $C$ aggregates the obfuscated $I_{proposed}(Z_{test}^B)$ from all testers, and filters out data with *negative* total score ($\sum_{\forall B} I_{proposed}(Z_{test}^B) < 0$).

The proposed influence offers many *advantages*. The designer may select any optimizer to perform the model updates, depending on the application at hand. We do not require the loss function to be twice differentiable and convex; only once differentiable. It is significantly more *computation and communication efficient*; an important prerequisite for any FL application. This is because agent $A$ only needs to send (a *small part* of) the model parameters $\theta$, and not his training data. Moreover, computing a few model updates (using e.g., SGD, or any other optimizer) is significantly faster than computing either the exact influence 1 or an approximation [20], due to the challenges mentioned above. Finally, and importantly, we ensure the *privacy* of both the train and test dataset of every agent.

### 3.3 Differentially Private Reporting of the Influence

We achieve this goal by obfuscating the influence reports using RAPPOR [13], which results in an $\varepsilon$-differential privacy guarantee [11]. The obfuscation process (permanent randomized response [33]) takes as input the agent's true value $v$ (binary) and privacy parameter $p$, and creates an obfuscated (noisy) reporting value $v'$, according to Equation 3. Subsequently, $v'$ is memorized and reused for all future reports on this distinct value $v$.

$$v' = \begin{cases} +1, & \text{with probability } \frac{1}{2}p \\ -1, & \text{with probability } \frac{1}{2}p \\ v, & \text{with probability } 1-p \end{cases} \tag{3}$$

$p$ is a *user-tunable* parameter that allows the agents themselves to *choose their desired level of privacy*, while maintaining reliable filtering. The worst-case privacy guarantee can be computed by each agent *a priori*, using the following formula [13]:

$$\varepsilon = 2\ln\left(\frac{1-\frac{1}{2}p}{\frac{1}{2}p}\right) \tag{4}$$

It is important to note that in a Federated Learning application, the center $C$ aggregates the influence sign from a *large number of agents*. This means that even under *really strict* privacy guarantees, *the aggregated influence signs (which is exactly what we use for filtering), will match the true value* in expectation. This results in *high quality filtering*, as we will demonstrate in Section 4.

The pseudo-code of the proposed approach is presented in Algorithm 1.

---

**Algorithm 1:** Filtering Poor Data Using Influence Approximation in Federated Learning

---

**1** $C$: The center ($C$) initializes the model $M_0(\theta_0)$
**2** **for** $t \in T$ *rounds of Federated Learning* **do**
**3** $\quad$ $C$: Broadcasts $\theta_t$
**4** $\quad$ **for** $Agent_i$ *in* $Agents$ **do**
**5** $\quad\quad$ $Agent_i$: Acts as a contributor ($A$). Performs $k$ local epochs with $Z_{A,t}$ on the partially-frozen model $\tilde{\theta}_t^A$.
**6** $\quad\quad$ $Agent_i$: Applies precisely crafted noise to $\tilde{\theta}_t^A$.
**7** $\quad\quad$ $Agent_i$: sends $\tilde{\theta}_t^A$ to $Agents_{-i}$.
**8** $\quad\quad$ **for** $Agent_j$ *in* $Agents_{-i}$ **do**
**9** $\quad\quad\quad$ $Agent_j$: Acts as a tester ($B$). Evaluates the loss of $Z_{test}^B$ on $\theta_t$
**10** $\quad\quad\quad$ $Agent_j$: Evaluates the loss of $Z_{test}^B$ on $\tilde{\theta}_t^A$
**11** $\quad\quad\quad$ $Agent_j$: Calculates vote $v$ (sign of influence), according to (Equation 2)
**12** $\quad\quad\quad$ $Agent_j$: Applies noise to $v$ according to his privacy parameter $p$ to get $v'$
**13** $\quad\quad\quad$ $Agent_j$: Sends $v'$ to $C$
**14** $\quad\quad$ $C$: Filters out $Agent_i$'s data based on the votes from $Agents_{-i}$ (i.e., if $\sum_{\forall B} I_{proposed}(Z_{test}^B) < 0$).
**15** $\quad$ $C$: Updates $\theta_t$ using data from unfiltered $Agents$;

---

# 4 Evaluation Results

In this section we report the results of a preliminary empirical evaluation of the proposed approach.

So far, we evaluated the method on two common datasets: MNIST and CIFAR 10. The corruption used for the evaluation is generated by applying a random label from the label space instead of the original label. For our experiments we corrupted 90% of the point per corrupted batch, while 30% of the total batches were corrupted.

1. **MNIST** Handwritten numerical digits [6]

2. **CIFAR10** Dataset of 32x32 colour images in 10 classes. [21]

## 4.1 Implementation

We used HuggingFace's implementation of Vision Transformers. [34] We opted to use Vision Transformer (ViT) for simplicity, and, importantly, because these models are on par with state of the art image classification models. [7] It is important to stress that our proposed influence approximation can be used with *any* gradient-descent based machine learning method.

The center $C$ provides a warm-up model, that has been trained for only a few epochs (3 in all our experiments). With the learning rate set to $2 \times 10^{-5}$, and regularization set to $10^{-2}$. This model keeps the best result unlike agent training, where we always take the final model.

Our evaluation involves a single round of Federated Learning. A small portion of every dataset (around 1%) was selected as the 'warm-up' data used by the center $C$ to train the initial model $M_0$. Each agent has two datasets: a training batch ($Z_A$, see Section 3.2, step (i)) which the agent uses to update the model when acting as the contributor agent, and a test dataset ($Z_{test}^B$, see Section 3.2, step (ii)), which the agent uses to estimate the sign of the influence when acting as a tester agent. The ratio of these datasets is $2 : 1$. The training batch size is 100 (i.e., the train dataset includes 100 points, and the test dataset 50 points). The learning rate for the agents has been increased compared

Table 1: Filtration performance metrics, with a 30% mislabel rate.

|  | Accuracy | Precision | Recall |
|---|---|---|---|
| MNIST | 100% | 100% | 100% |
| MNIST ($\varepsilon = 1$) | 100% | 100% | 100% |
| CIFAR10 | 100% | 100% | 100% |
| CIFAR10 ($\varepsilon = 1$) | 86.00% | 86.36% | 63% |

to the center model to $10^{-4}$, to emphasize the direction of model change. We used 100 agents. This means that each training batch was evaluated on $50 \times (100 - 1)$ test points, and that for each training batch (contributor agent $A$), the center collected (100-1) estimates on the influence sign (Equation 2). Finally, in a mislabeled batch, 90% of the labels have been assigned a random value from the label space.

### 4.2 Precision and Recall

Precision and recall are the most informative metrics to evaluate the efficiency of our filtering approach. Recall refers to the ratio of detected mislabeled batches aver all of the mislabeled batches. Meanwhile, precision represents the ratio of correctly identified mislabeled batches, over all batches identified as mislabeled. Table 1 shows that the proposed method performs well across all metrics, for both datasets, even under really strict privacy guarantees (i.e., $\varepsilon = 1$).

### 4.3 Privacy

Table 1 also shows the impact of the privacy guarantee on the achieved accuracy (note that $\varepsilon = 1$ is the privacy guarantee on both the training set, and the agent votes). We can see that there is of course a trade-off between privacy and efficiency of filtration. Yet, most importantly, our approach can provide high accuracy, even under *really strict, worst-case privacy requirements*. Importantly, our decentralized framework allows each agent to compute his *own* worst-case privacy guarantee *a priori*, using the Equation 4.

## 5 Conclusion

Privacy protection is a core element of Federated Learning. However, this privacy also means that it is significantly more difficult to ensure that the training data actually improve the model. Mislabeled, corrupted, or even malicious data can result in a strong degradation of the performance of model, and privacy protection makes it significantly more challenging to identify the cause.

In this work, we propose *'lazy' influence*, a *practical* approximation of the *influence* to obtain a meaningful score that characterizes the quality of training data and allows for effective filtering, while fully maintaining the privacy of both the train and test data under *strict, worst-case $\varepsilon$-differential* privacy guarantees.

The score can be used to filter bad data, recognize good and bad data providers, and pay data holders according to the quality of their contributions. We have documented empirically that poor data have a significant negative impact on the accuracy of the learned model, and that our filtering technique effectively mitigates this, even under strict privacy requirements $\varepsilon < 1$.

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
