# OpenReview forum: "Privacy-Preserving Data Filtering in Federated Learning Using Influence Approximation"
_NeurIPS.cc/2022/Workshop/Federated_Learning — FL-NeurIPS 2022 Poster_

### Official Review · Reviewer_1WbV · 2022-10-04
**The manuscript mainly studies how to evaluate the quality of local models in federated learning. The authors design a data filtering method based on a practical influence approximation in a privacy-preserving manner. The agent evaluates the others’ data by his own data and reports an obfuscated score based on differential privacy.**

The manuscript mainly studies how to evaluate the quality of local models in federated learning. The authors design a data filtering method based on a practical influence approximation in a privacy-preserving manner. The agent evaluates the others’ data by his own data and reports an obfuscated score based on differential privacy. Detailed comments are listed as follows.

1. Why only obfuscate data scores instead of model parameters?
2. The accuracy does not degrade significantly under really strong privacy guarantees is obvious. Because you do not perturb the model parameters, this does not prove the advantages of the scheme.
3. For each agent, how to use its own data to evaluate the influence of another agent’s data batch need to be in detail. Will this evaluation process significantly reduce the efficiency of FL?
4. In the simulations, why can the accuracy and recall rate of the MNIST and CIFAR10 datasets reach 100%?
5. Under really strict privacy guarantees, why the accuracy and recall of the MNIST dataset are not affected at all, but CIFAR10 is obviously affected?
6. In the experimental part, it is recommended to add the display of visualized results.

---

### Official Review · Reviewer_h3z6 · 2022-10-10
**Privacy-Preserving Data Filtering in Federated Learning Using Influence Approximation**

This paper presents a new algorithm to filter out the effects of
corrupted user data on a federated model during FL training.  Their
algorithm approximates the influence of model updates made by a user
by determining whether the updates (created by a configurable subset
of the user's data) lead to a better performing model for all users in
the federation in aggregate.  If a user's model updates are not
helpful for all users in aggregate, the updates are ignored by the
federation server during model update aggregation.  Furthermore, their
algorithm adds an additional layer of DP at each user.  Empirical
evaluation on MNIST and CIFAR10 show that the resulting models are
tolerant to corrupted data, and perform well.

I like the simplicity of the idea at its core.  However, there are
several issues that would be worth discussing in the paper.

While the algorithm is clever, it appears to make an IID assumption in
the federation, which may not be the case in practice.  Would your
algorithm be effective in non-IID FL settings?  If not, are there any
simple workarounds/fixes?

While the authors advocate retraining a subset of model parameters to
reduce network bandwidth consumption, the need to send every user's
model updates to all other users in the federation for data corruption
testing can easily run into scalability issues with even modestly
large federations with 100s of users.  How would you address this
issue in even larger federations?

The assumption that the federation server may have representative data
to bootstrap/initialize the model may not apply to some federations.
In such cases, the entire model may need to be trained at all users,
thereby exacerbating the network bandwidth consumption problem.

Instead of a table with the final numbers, graphs that show
performance at each training round will be more informative.
Furthermore, the authors should consider adding more data that
demonstrates filtering of users' model updates more directly.

Other aspects to consider adding in the paper: effect on model
convergence, empirical data on increase in network bandwidth and
compute consumption.

---

### Official Review · Reviewer_RUDf · 2022-10-17

This paper considers the data filtering problem in federated learning and how to combine it with differential privacy. The filtering process can be summarized as follows: 1) a worker pulls the public model from the server and trains on its local dataset, add noise to the model and send it to other workers; 2) other workers evaluate the received model on their local data distributions and compared the performance with the public model (influence); 3) workers add noise to the influence and send to the server; 4) the server aggregates all obfuscated influence values and decide to accept the updates or not. The influence function of samples is defined as the improvement of model quality with these samples compared to without. In this paper, they approximate the influence function with its sign.

- What is worrisome is that there lack of theoretical guarantees. For example, how can we ensure replacing influence with its sign lead to correct results? Besides, while it is not mentioned in the paper, the methods is limited to the IID case.
- The privacy claims need to be toned down. The differential privacy only preserves output privacy. There are orthogonal privacy concepts in federated learning, such as input privacy, which are not preserved in this paper.

---

### Decision · Program_Chairs · 2022-10-20

Accept (Poster)